# Accelerating exploration and representation learning with offline pre-training

**Bogdan Mazoure** [1]   **Jacob Bruce** [1]   **Doina Precup** [1]   **Rob Fergus** [1]   **Ankit Anand** [1]

## Abstract

Sequential decision-making agents struggle with long horizon tasks, since solving them requires multi-step reasoning. Most reinforcement learning (RL) algorithms address this challenge by improved credit assignment, introducing memory capability, altering the agent's intrinsic motivation (i.e. exploration) or its worldview (i.e. knowledge representation). Many of these components could be learned from offline data. In this work, we follow the hypothesis that exploration and representation learning can be improved by separately learning two different models from a single offline dataset. We show that learning a state representation using noise-contrastive estimation and a model of auxiliary reward separately from a single collection of human demonstrations can significantly improve the sample efficiency on the challenging NetHack benchmark. We also ablate various components of our experimental setting and highlight crucial insights.

## 1. Introduction

Sequential decision-making tasks with long horizon form an important class of problems, but can be notoriously difficult to solve (Bellemare et al., 2013; Vinyals et al., 2019; Küttler et al., 2020). For example, NetHack is a popular terminal-based rogue-like video game with a multitude of sub-tasks needing to be completed in order to win. So far, only around 1% of human players can fully solve the game (Hambro et al., 2022). How does an agent prioritize conducting an invocation over finding a dungeon key, or fully explore a maze? Undirected exploration is hopeless in this scenario, since the agent has to try out a number of actions exponential in trajectory length, quickly becoming intractable even for simple sub-tasks. However, as recent works(Anonymous, 2023) show, even more powerful exploration techniques

such as Burda et al. (RND, 2018) do not make significant progress on harder tasks within the NetHack Learning Environment, or NLE (Küttler et al., 2020).

Often, this problem is addressed by determining the action expansion order through either modifying the agent's intrinsic motivation (i.e., exploration, Dudik et al., 2011; Bellemare et al., 2016; Burda et al., 2018; Ecoffet et al., 2019; Guo et al., 2022) or structuring its internal world representation (i.e., representation learning, Jaderberg et al., 2016; Anand et al., 2019; Srinivas et al., 2020; Mazoure et al., 2020; 2021; Eysenbach et al., 2022). Both exploration and representation learning approaches to long horizon RL problems have seen their respective successes in complex domains e.g. Atari's Montezuma's Revenge (Ecoffet et al., 2019) and maze-like environments (Raileanu & Rocktäschel, 2020). Some attempts were made to combine both exploration and representation learning (Misra et al., 2020; Yarats et al., 2021), but they are limited to specific families of MDP (e.g. BlockMDP) which might not hold in complex real-world scenarios.

How can one leverage the representation learning paradigm in order to achieve more sample-efficient exploration in long-horizon RL problems? While exploration cannot be performed without online interactions, it is possible to amortize the sample complexity of representation learning by pre-training the agent on offline data. This approach of offline pre-training followed by online fine-tuning has lead to important advances in complex, multi-modal control tasks such as Minecraft (Fan et al., 2022; Baker et al., 2022). In these approaches, human curators were used as a labeling mechanism on incomplete trajectory data, which was then used in the representation learning phase. Does executing classical exploration strategies on top of state representations with *desirable* properties (i.e., capturing the progress of human demonstrations towards solving the downstream task) make them more sample-efficient?

In this work, we posit the hypothesis that combining representation learning with exploration improves performance on long-horizon tasks. We test our hypothesis in the NetHack Learning Environment (Küttler et al., 2020), a challenging domain with high-dimensional state and action spaces, strong notion of forward progress and organic definition of multiple subgoal tasks. While algorithms based on a combination of RL and imitation learning can solve easier

---

[1]Google DeepMind. Correspondence to: Bogdan Mazoure (now at Apple) <bogdan.mazoure@apple.com>.

Interactive Learning with Implicit Human Feedback Workshop at ICML 2023.

subtasks such as the NeurIPS'21 challenge, completing the game is only possible for symbolic agents or human experts. Therefore, NLE provides an excellent opportunity bridging the performance gap between RL agents and humans, unlike other common benchmarks such as the Arcade Learning Environment (Bellemare et al., 2013).

Previous work, Explore-Like-Exports (ELE, Anonymous, 2023) showed that many of the sparse reward tasks in NetHack can be solved by learning a simple scalar function which predicts the expert progress or temporal distance between two observations in any trajectory from the expert data. While sparser tasks are solved by introducing this additional reward, the performance on dense reward tasks like NeurIPS' 21 challenge and scout suffer in comparison to baseline. Secondly, we also hypothesize that ELE discards useful information contained in expert trajectories by compressing the dataset into a single scalar-valued function. We address these issues by using the same expert data to learn representations by contrastive pre-training and in conjunction with ELE, develop an agent that not only has significantly better sample efficiency but also improved performance than ELE and standard imitation learning baselines on a spectrum of tasks ranging from sparse to dense in a challenging domain like NetHack. This work illustrates how one can use the same dataset to learn representations and auxiliary reward, thereby achieving better sample efficiency and performance.

Specifically, we show that a simple offline pre-training scheme based on contrastive learning (Eysenbach et al., 2022; Mazoure et al., 2022) can be used in conjunction with ELE i.e learning a progress reward (Anonymous, 2023) on the same expert data. This not only improves the sample efficiency and base performance of Muesli, a strong RL baseline (Hessel et al., 2021a) but also using representation learning or progress reward alone on NetHack in a wide variety of tasks.

## 2. Related works

### 2.1. Auxiliary tasks in RL

While one of the main goals in RL problems is to find a policy which maximizes expected reward, it can often be challenging due to a multitude of factors, e.g. sparse reward signal, untractably large policy space, long task horizon, etc. Since the problem is extremely hard in its current formulation, it is possible to augment it with external learning signals, which are notably specified via auxiliary downstream tasks. Auxiliary learning objectives have been widely studied in the literature, in both online (Jaderberg et al., 2016; Stooke et al., 2021) and offline settings (Schwarzer et al., 2021; Yang & Nachum, 2021). They can be used to equip RL agents with desirable inductive biases, e.g. disentanglement (Higgins et al., 2017), alignment and uniformity (Wang & Isola, 2020) or predictivity of future observations (Jaderberg et al., 2016;

Mazoure et al., 2020).

World models provide one natural pre-training objective for RL agents, allowing it to capture crucial parameters of the environment such as transition dynamics, reward function and initial state distribution. Single-step world models such as DreamerV3 (Hafner et al., 2023) and Director (Hafner et al., 2022) equip RL agents with single-step transition and reward models that can then be used for planning. However, training such models from offline data is non-trivial and costly; using them in online settings is computationally inefficient as it requires unrolling the sequence of latent states and actions in an autoregressive manner. On the other hand, infinite-horizon models such as $\gamma$-models (Janner et al., 2020) or contrastive value functions (Eysenbach et al., 2022; Mazoure et al., 2022) are harder to learn, but directly capture the probability of observing a future state when rolling out from the current state.

### 2.2. Exploration

Some of the inductive biases for challenging tasks can be learned from offline demonstrations, e.g. human interactions with the environment (Reid et al., 2022; Fan et al., 2022; Baker et al., 2022). In hard tasks with sparse rewards and long horizons, agents need to rely on other forms of supervision, i.e. intrinsic motivation. Intrinsic motivation for guided exploration has been an active area of research in the past years, encompassing count-based exploration (Bellemare et al., 2016; Tang et al., 2017), knowledge gathering (Kim et al., 2018; Zhang et al., 2021) and curiosity (Burda et al., 2018; Raileanu & Rocktäschel, 2020). However, curiosity-based exploration from tabula rasa is still a hard problem in some tasks (e.g. NetHack), and hence warrants the use of learned auxiliary rewards from data.

### 2.3. Learning from demonstrations

In domains where RL agents have not yet achieved human-level performance, learning can be accelerated by training on demonstrations of experts (symbolic agents, humans, etc). Classical imitation learning methods like Behavior Cloning (Pomerleau, 1988), is one of the most effective and popularly used methods in presence of large quantities of data in complex domains like Minecraft(Baker et al., 2022), computer control(Humphreys et al., 2022) etc. Other approaches like GAIL (Ho & Ermon, 2016) learns a discriminator to distinguish expert trajectories from agent trajectories which could be modeled as a reward. These methods have been further extended to work on expert trajectories without actions as BCO (Torabi et al., 2018a) and GAIfO (Torabi et al., 2018b). Another generative approach FORM (Jaegle et al., 2021) augments the environment reward by an additional reward by learning a forward generative model of transition dynamics from offline data and rewarding transitions under the learned model. In scenarios of unlabeled(that contain

*Figure 1.* Experimental setting studied in this work. An offline pre-training phase of the agent's representations $f$ as well as the progress model $g$ (top) is followed by an online fine-tuning phase, during which the agent uses both $f$ and $g$ to collect information about the environment.

no actions) datasets like NetHack , experts can be used to annotate existing datasets without action information, e.g. add action information based on external sources such as in MineDojo or VPT (Fan et al., 2022; Baker et al., 2022). However, such labeling schemes can involve collecting data from human experts or training complex RL agents from scratch, both of which are prohibitively expensive in many scenarios. Alternatively, demonstrations can be used to guide RL agents through intrinsic motivation using learned heuristic functions. For example, ELE (Anonymous, 2023) learns a heuristic function quantifying temporal progress in expert trajectories. It outperforms prior state-of-the-art on 7 NetHack tasks with sparse rewards, but still does not solve the game itself. We hypothesize that the main drawback of ELE is that it reduces the pre-training dataset to a single scalar-valued function, and does not extract the most information out of the data. Specifically, the degree to which a dataset heuristic is beneficial for a given online task depends on its alignment with the optimal value function in that MDP (Cheng et al., 2021). In this work, we focus on using offline data which does not contain any actions and hence, limit ourselves to compare with ELE, BCO, GAIfO and FORM as standard imitation learning baselines.

## 3. Preliminaries

### 3.1. Reinforcement learning

The classical reinforcement learning setting assumes that the environment follows a Markov decision process $M$ defined by the tuple $M = \langle \mathcal{S}, S_0, \mathcal{A}, \mathcal{T}, r, \gamma \rangle$, where $\mathcal{S}$ is the state space, $\mathbb{P}[S_0], S_0 \in \mathcal{S}$ is the distribution of starting states, $\mathcal{A}$ is the action space, $\mathcal{T} = \mathbb{P}[\cdot|s_t, a_t]: \mathcal{S} \times \mathcal{A} \to \Delta(\mathcal{S})$ is the transition kernel[1], $r: \mathcal{S} \times \mathcal{A} \to [r_{\min}, r_{\max}]$ is the reward function and $\gamma \in [0,1)$ is a discount factor. The environment is initialized in $s_0 \sim \mathbb{P}[S_0]$. At every timestep $t = 1,2,3,..$, the policy $\pi: \mathcal{S} \to \Delta(\mathcal{A})$, samples an action $a_t \sim \pi(\cdot|s_t)$. The environment then transitions into the next state $s_{t+1} \sim \mathcal{T}(\cdot|s_t, a_t)$ and emits a reward $r_t = r(s_t, a_t)$. The state value function is defined as the cumulative per-timestep discounted rewards

---

[1] $\Delta(\mathcal{X})$ denotes the entire set of distributions over the space $\mathcal{X}$.

collected by policy $\pi$ over an episode of length $H$:

$$V^\pi(s_t) = \mathbb{E}_{\mathbb{P}^\pi_{t:H}} \left[ \sum_{k=0}^{H-t} \gamma^k r(s_{t+k}, a_{t+k}) | s_t \right], \quad (1)$$

where $\mathbb{P}^\pi_{t:t+K}$ denotes the joint distribution of $\{s_{t+k}, a_{t+k}\}_{k=1}^K$ obtained by deploying $\pi$ in the environment $M$ from timestep $t$ to timestep $t + K$. The state-action value function is defined analogously as

$$Q^\pi(s_t, a_t) = \mathbb{E}_{\mathbb{P}^\pi_{t:H}} \left[ \sum_{k=0}^{H-t} \gamma^k r(s_{t+k}, a_{t+k}) | s_t, a_t \right], \quad (2)$$

such that $Q^\pi(s_t, a_t) = r(s_t, a_t) + \gamma \mathbb{E}_{\mathcal{T}(s_t, a_t)}[V^\pi(s_{t+1})]$.

The reinforcement learning problem consists in finding a Markovian policy $\pi^*$ that maximizes the state value function over the set of initial states:

$$\pi^* = \max_{\pi \in \Pi} \mathbb{E}_{\mathbb{P}[S_0]}[V^\pi(s_0)], \quad (3)$$

for $s_0 \sim \mathbb{P}[S_0]$ and set of policies $\Pi$. Alternatively, the value function can also be re-written as the expectation of the reward over the geometric mixture of $k$-step forward transition probabilities:

$$V^\pi(s_t) = \frac{1}{1-\gamma} \mathbb{E}_{(s,a \sim \rho^\pi(s_t) \times \pi(s))}[r(s,a)], \quad (4)$$

where

$$\rho^\pi(s|s_t) = (1-\gamma) \sum_{\Delta t=1}^{H} \gamma^{\Delta t-1} \mathbb{P}[S_{t+\Delta t} = s|s_t; \pi]$$

$$= \mathbb{E}_{\Delta t \sim \text{Geo}_t^H(1-\gamma)}[\mathbb{P}[S_{t+\Delta t}|s_t, \Delta t; \pi]], \quad (5)$$

and $\text{Geo}_t^H(1-\gamma)$ denotes a truncated geometric distribution with probability mass re-distributed over the interval $[t, H]$.

This decomposition of the value function is useful in scenarios in which the environment's rewards are delayed, leaving the learner only with access to states. It has been used in previous works based on the successor representation (Dayan, 1993; Barreto et al., 2016) and, more recently, explicit (Janner et al., 2020) and implicit (Eysenbach et al., 2022; Mazoure et al., 2022) infinite-horizon models.

## 3.2. Explore-Like-Experts

Exploration in long-horizon problems with large action spaces and sparse rewards is hard: an uninformed agent would have to try $|\mathcal{A}|^H$ actions for a horizon length $H$, which is infeasible in NetHack, where $|\mathcal{A}| = 121$ and $H$ can be close to $10^6$ (Küttler et al., 2020). Augmenting the uninformative extrinsic reward with an intrinsic signal which drives the agent to visit rare state-action pairs can directly translate into higher overall returns, as the learner uncovers more of the extrinsic reward structure. More formally, it is achieved by constructing an auxiliary MDP $M'$, where the reward function at timestep $t$ is a combination of the extrinsic reward from $M$ as well as some heuristic function $h : \mathcal{S} \times \mathcal{A} \to \mathbb{R}$:

$$r'(s_t, a_t) := r(s_t, a_t) + \lambda h(s_{0:t}, a_{0:t}) \tag{6}$$

When $h(s_{0:t}, a_{0:t}) = \gamma \mathbb{E}_{\mathcal{T}(s_t, a_t)}[v(s_{t+1})]$ for some function $v : \mathcal{S} \to \mathbb{R}$, then solving Equation (3) in $M'$ is equivalent to finding $\pi$ in $M$ if $v$ approximates $V^*$ (original value function), but with a lower discount factor $\gamma' = \gamma(1 - \lambda)$ when $\lambda < 1$ (Cheng et al., 2021)[2]. While setting $v = V^*$ leads to maximal sample efficiency, $V^*$ is not accessible in practice, and $h$ has to be selected based on the structure of $M$. In particular, good heuristics can be constructed from data using existing pessimistic offline RL algorithms, e.g. improvable heuristics lead to small estimation bias of $V^*$, since they are smaller than the maximum of the Bellman backup.

Intuitively, given two states, how to prioritize one over the other during the exploration process? If we had a systematic way to evaluate which state is closer to the goal under the optimal policy, then we could force the agent to expand that state during the exploration phase. Maximizing the progress in the task can be captured through a monotonically increasing function of states learned from optimal data (where, by definition, progress is maximal). Specifically, the Explore Like Experts algorithm (ELE) (Anonymous, 2023), first trains a function $g : \mathcal{S} \to \mathbb{R}$ by solving

$$g^* = \min_{g \in \mathcal{F}} \mathbb{E}_{\mathcal{D}}[\ell_{\mathrm{ELE}}(g, s_t, s_{t+\Delta t})] \tag{7}$$

where

$$\ell_{\mathrm{ELE}}(g, s_t, s_{t+\Delta t}) = \{g(s_t, s_{t+\Delta t}) - \mathrm{sgn}(\Delta t)\log(1 + |\Delta t|)\}^2, \tag{8}$$

$\mathcal{D}$ is a set of expert human demonstrations and $\Delta t \sim \mathrm{LogUniform}(0, 10^4)$. Specifically, Equation (8) does mean-squared error regression in the signed log-space to predict the time offset $\Delta t$ from states $s_t$ and $s_{t+\Delta t}$.

In the second step, ELE uses the pre-trained progress model in place of the $h$ heuristic in Equation (6)

$$r^{\mathrm{ELE}}(s_t, a_t) := r(s_t, a_t) + \lambda g(s_{t-\Delta t}, s_t), \tag{9}$$

---

[2]If $v$ is an improvable heuristic aligned with the optimal value function, then the discount factor in $M'$ is lowered.

an approximation of the local progress from $s_{t-\Delta t}$ to $s_t$. While $\Delta t$ was sampled by LogUniform distribution while training $g$, it was kept fixed during the online phase in ELE. In other words, auxiliary reward always computed progress with respect to a state $\Delta t$ steps behind from current state.

## 3.3. Contrastive representation learning

The conditional probability distribution of $s_{t+\Delta t}$ given $s_t$ can be efficiently estimated using an implicit model $f : \mathcal{S} \times \mathcal{S} \to \mathbb{R}$ trained via contrastive learning (Oord et al., 2018) on offline demonstrations $\mathcal{D}$ by solving:

$$f^* = \min_{f \in \mathcal{F}} \mathbb{E}_{\mathcal{D}}[\ell_{\mathrm{Contrastive}}(f, s_t, s_t^+, s_t^-)] \tag{10}$$

where

$$\ell_{\mathrm{Contrastive}}(f, s_t, s_t^+, s_t^-) = -\log \frac{e^{f(s_t, s_t^+)}}{\sum\limits_{s_t' \in s_t^+ \cup s_t^-} e^{f(s_t, s_t')}} . \tag{11}$$

To approximate the occupancy measure defined in Equation (5), positive samples are sampled from $s_t^+ \in \{s_{t+\Delta t}; \Delta t \sim \mathrm{Geo}_t^H(1-\gamma)\}$ for timestep $t$. Specifically, they are constructed by first sampling the interval $\Delta t$ from $\mathrm{Geo}_t^H(1-\gamma)$ and subsequently querying $s_{t+\Delta t}$ in the same episode. The negative samples $s_t^-$ are uniformly sampled from any timestep within the current or any other episode.

Minimizing Equation (11) over $\mathcal{D}$ yields a function $f^*$ which, at optimality, approximates the future state visitation probability under $\pi$ up to a multiplicative term (Ma & Collins, 2018; Poole et al., 2019).

$$f^*(s_t, s_{t+\Delta t}) \propto \log \frac{\mathbb{P}[s_{t+\Delta t} | s_t; \pi]}{\mathbb{P}[s_{t+\Delta t}; \pi]} . \tag{12}$$

It should be noted that the time offsets in both ELE's progress model and in the contrastive pre-training phase are sampled from similar distributions (see Appendix A.1). In the following section, we show how $f$ can be used for accelerating exploration in the online setting.

## 4. Methodology

In this section, we provide details of both the pre-training phase on offline human demonstrations and using these state representations in Muesli (Hessel et al., 2021a), a strong RL baseline. We also describe how to use same offline data for training progress model as well as training ELE's progress model.

**Pre-training state representations with contrastive training** The idea behind pre-training offline representations is fairly straightforward: learn fundamental inductive biases required for exploration from existing demonstrations (e.g. state connectivity structure, action effects, perceptual local invariance, sequential dependence of states), therefore improving the sample-efficiency of the agent during the online phase.

Figure 1 and Algorithm 1 (see Appendix) outline the general paradigm of offline pre-training with online learning used in all of our experiments, and which relies on finding $\phi$ that minimizes Equation (11) over the set of possible encoders. The pre-trained encoder is kept frozen or fixed through out the training so that even if agent explores a new part of state space, it does not drift away from the pre-trained representation. But it should be noted that we use a standard LSTM and MLP on top of the frozen encoder which are trained through out the training. In our experiments, we observe that these pre-trained representations themselves are very useful for improving sample efficiency of dense tasks but fail to solve the sparse version of tasks themselves in NetHack where a single reward is provided in the whole episode(more details on sparsity of tasks in experimental section). To address this, we add the ELE progress reward (Anonymous, 2023) learnt from the same data to the environment reward. The main hypothesis is that the signal from the progress model solves the problem of hard exploration in sparse reward tasks while pre-training helps for faster learning and hence, improving sample efficiency. Hence, one can use the same dataset to learn signals of representation as well as additional reward to assist exploration providing orthogonal benefits.

**Why contrastive pre-training?**     Why should one pick the contrastive pre-training scheme over any other objective? First, as mentioned in Section 5, we test our hypothesis on NetHack data containing only sequences of states (i.e. no actions nor rewards), which prevents the use of inverse RL and reward prediction objectives such as SGI (Schwarzer et al., 2021). Second, strong empirical evidence from prior works suggests that contrastive learning is optimal for value function representation and outperforms latent and reconstruction based forward losses (Eysenbach et al., 2022). Finally, latent-space prediction losses are known to be prone to representation collapse, i.e. when bootstrapped prediction targets are being exactly matched by the online network and become independent of the state (Gelada et al., 2019). Contrastive learning avoids representation collapse due to the presence of negative samples which ensure uniformity of coverage on the unit sphere (Wang & Isola, 2020).

## 5. Experiments

In this section, we conduct a series of experiments to validate our central hypothesis: combining representation learning with exploration improves the agent's sample complexity more than representation learning or exploration by themselves.

### 5.1. Experimental Details

**Baselines:**    Our main results are obtained by comparing the performance of *tabula rasa* Muesli and ELE (Anonymous,

2023) with their counterparts using pre-trained state representations. In addition, we also compare with standard baselines that use action-free expert demonstrations: GAIfO (Torabi et al., 2018b), BCO (Torabi et al., 2018a) and FORM (Jaegle et al., 2021), the same baselines as ELE. All these baselines learn from the same offline data and are implemented on top of Muesli agent (Hessel et al., 2021a) for fair comparison and we use the same hyperparameters as provided in ELE (Anonymous, 2023). Since previous work (Eysenbach et al., 2022) demonstrated that contrastive pre-training performs much better than other representation learning techniques, we limit our comparison in this work to contrastive pre-training. We performed preliminary investigations for other types of pre-training using forward prediction and latent models but they performed inferior to contrastive pre-training. The motivation for using contrastive pre-training of state representations is two-fold: 1) it allows Muesli to predict value functions using a linear layer, making the task simpler, and 2) it was shown to perform significantly better than latent-space or reconstruction-based objectives (see Eysenbach et al., 2022).

**Tasks**    We use 7 different tasks from NetHack ranging from dense to sparse rewards which were proposed in (Küttler et al., 2020) and ELE (Anonymous, 2023). On the dense side of the spectrum, we use the **Score** and **Scout** tasks, which reward the agent for increasing the game score and revealing tiles in the dungeon, respectively. At the sparse end, the **Depth N**, **Level N**, and **Oracle** tasks deliver a reward of zero at all timesteps until a trigger condition is met: reaching a particular dungeon depth, achieving a particular experience level, or finding and standing next to the Oracle character (found between dungeon depths 5 and 9 inclusive); these sparse tasks terminate when the condition is met. We believe that the wide range of sparsity levels exhibited by this task collection represents a good selection of conditions under which to evaluate the sample complexity of the algorithms we compare.

**Dataset**    We use the NAO Top 10 dataset proposed in previous work (Anonymous, 2023) which consists of human games from top 10 players on nethack.alt.org. These trajectories are useful for pre-training our contrastive representations as this dataset provides a good balance of trajectory quality and diversity(Anonymous, 2023) to learn representations. This dataset consists of approximately 16K trajectories of expert play, with a total of 184M transitions.

**Frame Budget**    As we want to compare algorithms on sample efficiency, we use 200M actor steps inspired from the Atari benchmark (Bellemare et al., 2013) on all these tasks, with the exception of the **Oracle** task. As this task poses a significantly harder exploration challenge, we allow a larger budget of 500M actor steps.

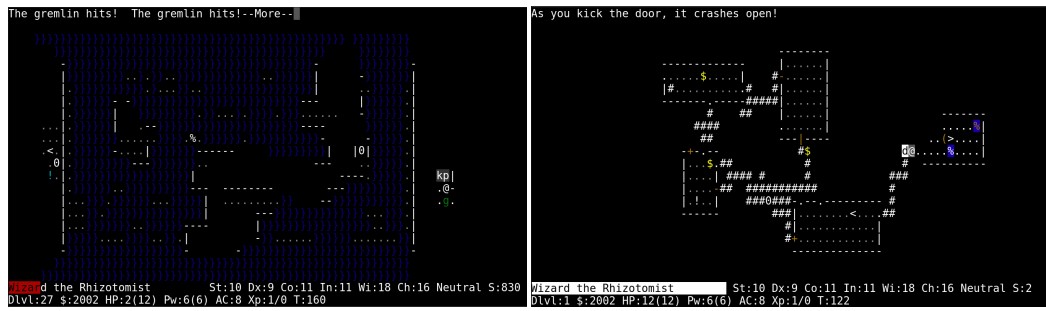

*Figure 2.* Examples of observations generated by the NetHack Learning environment.

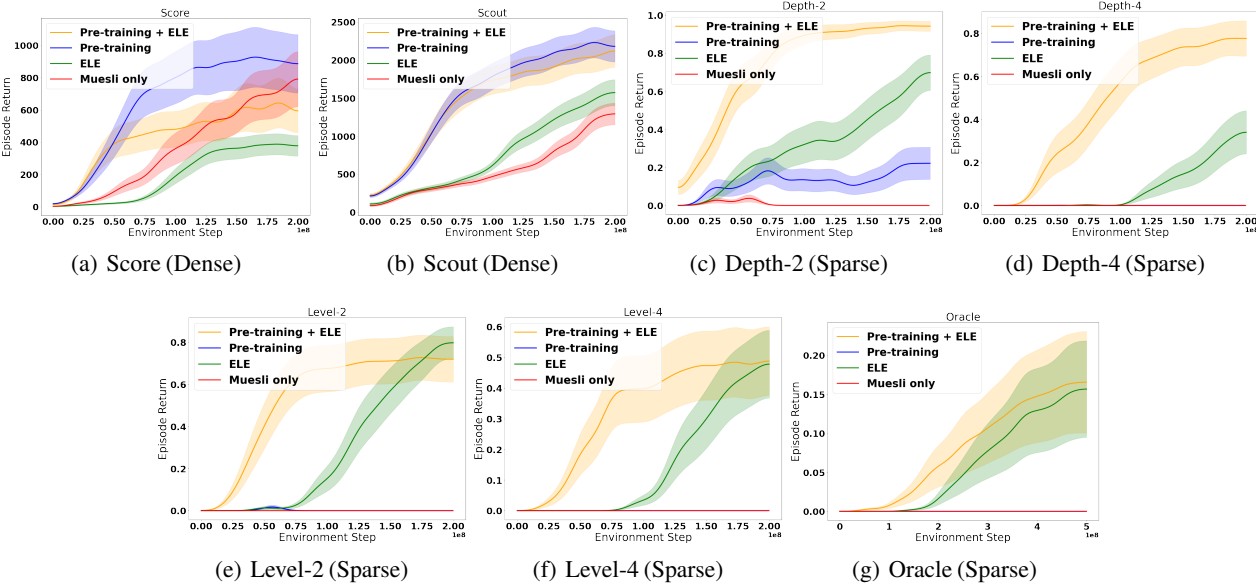

(a) Score (Dense)  (b) Scout (Dense)  (c) Depth-2 (Sparse)  (d) Depth-4 (Sparse)

(e) Level-2 (Sparse)  (f) Level-4 (Sparse)  (g) Oracle (Sparse)

*Figure 3.* Episode returns of Muesli and ELE, with and without pre-trained state representations. Dense reward tasks like Score and Scout benefit immensely from contrastive pre-training in both performance and sample efficiency. While ELE's exploration reward is needed to solve sparse reward tasks, contrastive pre-training augments ELE by improving sample efficiency for all the sparse tasks. All curves are reported over 5 random seeds ± one standard deviation.

**Architecture**   Inspired from previous work (Anonymous, 2023), we use a Residual Network (ResNet, He et al., 2016) architecture which encodes $80 \times 24$ TTY arrays (shown in Figure 2) with a series of 2d convolutional block. This model acts as encoder which is used by contrastive pre-training, ELE's progress model as well as Muesli agent. During online phase, we pass the generated representation with a recurrent network (LSTM) and MLP to predict policy and value heads in Muesli. In case of contrastive pre-training, we simply pass the ResNet encoder through an MLP in order to project the state features into a latent space. The ELE's progress model fuses the two states given as inputs, which are then passed through a similar ResNet followed by an MLP to predict a scalar value in logarithmic space, that corresponds to the temporal distance between both input states.

## 5.2. Results

We state the main results followed by ablation for different components. All our experimental results are ran with 5 random seeds and are plotted with ± standard deviation.

**Comparison of Progress Model and baseline with and without Pre-training**   Figure 3 shows that equipping strong RL algorithms such as Muesli and ELE with human demonstrations via offline pre-training significantly improves the sample complexity of the underlying method. While ELE significantly outperformed Muesli on the sparser tasks, the performance did not improve on the denser **Score** and **Scout** tasks and in fact was inferior to Muesli. Using contrastive pre-training with both Muesli and ELE, however, significantly improves its performance in the sample regime under investigation in this work. On the sparse tasks **Depth**

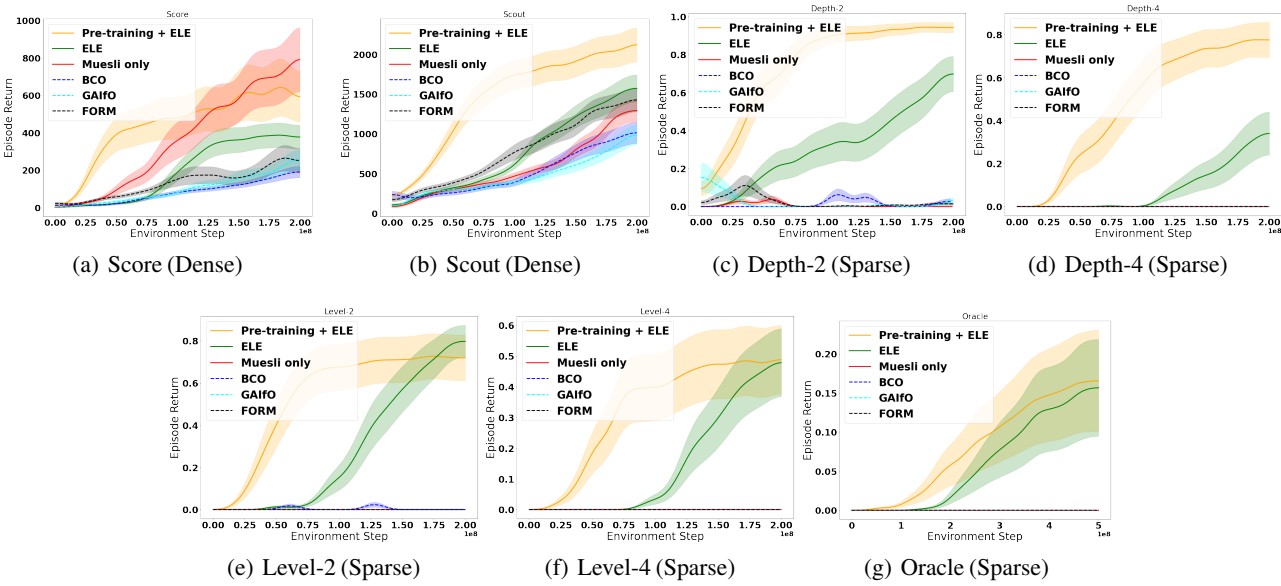

*Figure 4.* Episode returns of ELE with pre-trained representation in contrast to standard imitation learning baselines (without access to actions) like BCO, GAIfO and FORM. Contrastive Pre-training + ELE outperforms all baselines on sparse as well as dense tasks. All curves are reported over 5 random seeds ± one standard deviation.

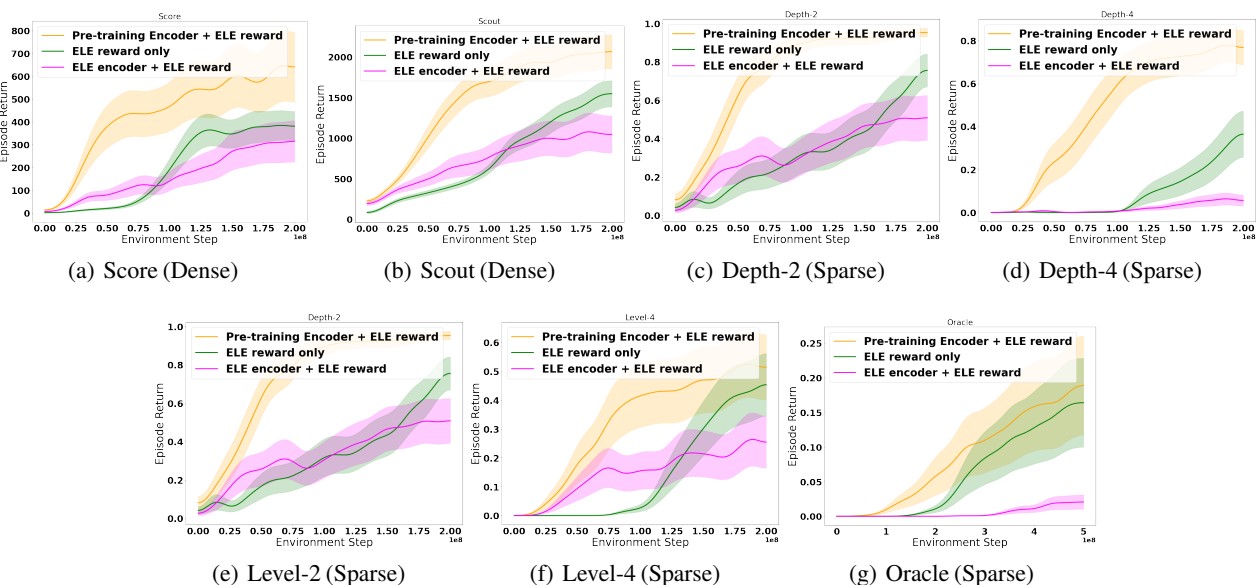

*Figure 5.* Episode returns of encoder extracted from ELE to Encoder separately trained by contrastive pre-training. All the tasks demonstrate that training a separate encoder by contrastive pre-training is much more useful on all the tasks illustrating ELE and contrastive pre-training capture two different dimensions of exploration and representation learning respectively.

**2**, **Depth 4**, **Level 2**, **Level 4** and **Oracle**, pre-training with ELE significantly improves performance in the low sample regime. It should be noted that the contrastive pre-training without an exploration bonus struggles to solve the sparser tasks, and the progress reward is clearly beneficial in this case. This illustrates that the same dataset can be used for both pre-training representations as well as learning an exploration reward, and that these two different applications of human data target orthogonal problems: exploration bonuses help the agent discover the reward, and representation learning improves its ability to exploit it.

**Comparison with Standard Imitation Learning Baselines:** Figure 4 show the comparison of ELE + Pre-training with other standard imitation learning baselines like GAIL from Observations (GAIfO) (Torabi et al., 2018b), Behavior Cloning from Observations (BCO) (Torabi et al., 2018a) and FORM (Jaegle et al., 2021). On the sparser tasks, only ELE and ELE + Pre-training are able to solve them at all, and contrastive pre-training improves convergence speed significantly. Dense tasks like **Score** and **Scout** are learned by many of the baselines, but contrastive pre-training significantly improves sample efficiency.

### 5.3. Ablations

**Is using progress model's representation encoder as good as contrastive pre-training?** An interesting question which stems from this work is that if the ELE's progress model is useful for exploration, could we use the trained progress model's torso as an encoder for initializing representations as well ? We experiment with extracting the torso of trained progress model and use it to initialize the representation encoder (instead of using encoder from contrastive pre-training). Figure 5 shows the comparison of using progress model representations v/s training these separately by contrastive pre-training on the same data. We observe that using ELE's torso as encoder and additional reward from ELE takes off faster but eventually achieves poor performance than ELE itself and is significantly poor than using ELE with contrastive pre-training encoder.

**Do pre-trained state representations need to be finetuned during the online phase?** Next, we study the effect of freezing[3] the pre-trained representations of observation encoder from pre-training phase. We observe no significant difference on most tasks with or without freezing representations. Figure 6 shows 1 sparse task and 1 dense task for this ablation (more details in appendix). We stick with freezing representations for the online phase as our default setting through out the paper.

**How does encoder architecture impact its performance?** A natural question which arises when pre-training the state representations offline is how well does the model capture the future states. In all of our experiments, we use a simple ResNet (He et al., 2016) which takes as inputs an array of TTY characters. However, recent works have shown that the local invariance biases from the 2d convolutions can be learned through a vision transformer model (ViT, Dosovitskiy et al., 2020; Raghu et al., 2021), which positioned ViT as a competitive alternative to standard convolution-based architectures. The main drawback of ViTs is their need to be trained on vast amounts of data, which is abundant in NetHack. We have conducted pre-training experiments comparing the contrastive

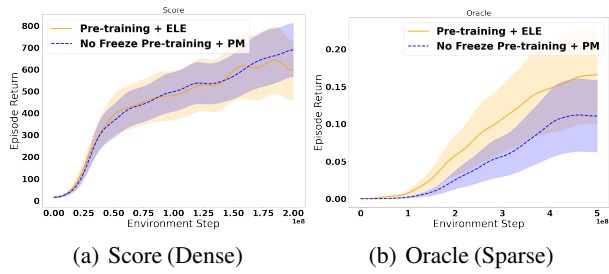

(a) Score (Dense)  (b) Oracle (Sparse)

*Figure 6.* Episode returns of with/without freezing representations of contrastive pre-training during online phase. There is no significant difference between freezing or not freezing the state representations. We show 1 dense task (Score) and 1 sparse task (Oracle) as representatives. All other tasks are shown in the Appendix

prediction accuracy of convolution-based models with that of ViTs. Results shown in Figure 7 hint that ResNet-like models are better suited for NLE, as they obtain better training set and test set categorical accuracy as compared to ViTs.

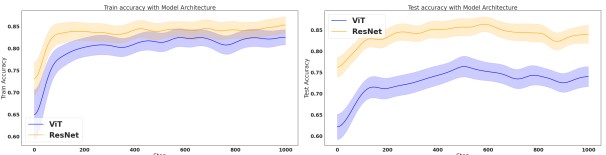

(a) State prediction accuracy on the training episodes.  (b) State prediction accuracy on the test episodes.

*Figure 7.* Ablation on different Model Architectures for contrastive pre-training. We observe that Vision transformer performs much worse than ResNet architecture.

## 6. Discussion

In this work, we posited that same offline data could be used for learning representations as well as learning an auxiliary reward to aid exploration and training these models separately provide orthogonal benefits. We show that pre-training state representations using contrastive learning and then using this network to initialize the representations provides a large sample-efficiency improvement. However, using pre-training alone fail to solve sparse tasks. We address the problem by adding a learned auxiliary reward and observe that pre-training helps in representation learning and auxiliary reward aids exploration. We validate our hypothesis in the NetHack, a challenging rogue-like terminal-based game with large state and action spaces, long task horizon and strong notion of forward progress.

---

[3]Fixing a set of weights during the online phase.

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

# A. Appendix

## A.1. Experimental details

**Distribution of $\Delta t$.** For both the contrastive pre-training scheme, as well as ELE's progress model, the distribution of offsets between current and future states has a similar shape, albeit not being exactly equal: our current work chooses $\Delta t \sim \text{Geom}_t^H(1-\gamma)$, which has three parameters, while ELE uses $\Delta t \sim \text{LogUniform}(t, H)$. The advantage of using a truncated geometric distribution is two-fold: 1) the support is discrete and therefore allows to directly sample time offsets without truncation, and 2) the extra parameter $\gamma$ is directly tied to the environment and allows to control for the shape of the distribution, making the approach more flexible. Specifically, increasing $\gamma \to 1$ brings the distribution closer to $\text{Uniform}(t, H)$, while decreasing $\gamma \to 0$ brings it closer to $\delta(t)$.

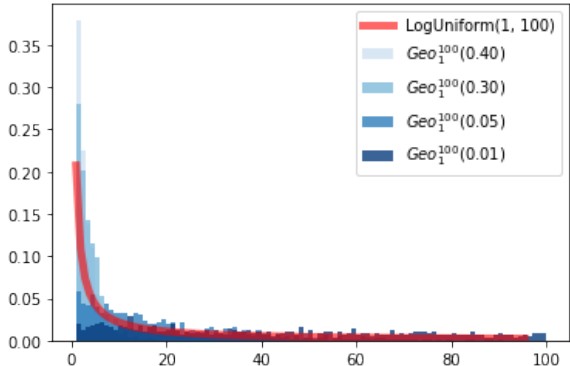

*Figure 8.* Comparison of a log-uniform distribution with truncated geometric distributions with various success probabilities. It can be seen that both coincide for some success probability.

**Computational resources** We ran all of our experiments on 8 V100 GPUs for 6 hours.

## A.2. Additional results

**Freezing the pre-trained representation.** Figure 9 shows the performance difference between using a frozen pre-trained state representation in the online phase, vs allowing the gradient of the online loss propagate through the representation.

## A.3. Algorithm

Algorithm pseudocode is provided in Algorithm 1.

---

**Algorithm 1:** Pre-training with online learning

---

**Input** : Dataset $\mathcal{D}^\mu \sim \mathbb{P}_{0:H}^\mu$, environment $M$, state embedding $\phi$, policy $\pi$

```
/* Pre-train φ using Equation (11)                                    */
```
1 **for** *minibatch* $\mathcal{B} \sim \mathcal{D}^\mu$ **do**
2     Update $\phi$ using $\nabla_\phi \ell_{\text{Contrastive}}(\phi)(\mathcal{B})$ ;
```
/* Learn forward progress model (ELE)                                 */
```
3 **for** *minibatch* $\mathcal{B} \sim \mathcal{D}^\mu$ **do**
4     Update $g$ to minimize Equation (8) ;
5 Initialize the online encoder with $\phi$;
6 **for** *iteration* $i = 1 \to N$ **do**
7     Generate the trajectory $\tau$ using acting policy $\pi(\theta)$
```
    /* Update reward in τ by adding progress reward                  */
```
8     **for** $(s_t, a_t, s_{t+1}, r_t)$ *in* $\tau$ **do**
9        Update $r_t \leftarrow r_t + \lambda g(s_{t-k}, s_t)$
10     Update $\theta$ using Muesli Loss function keeping $\phi$ frozen.

---

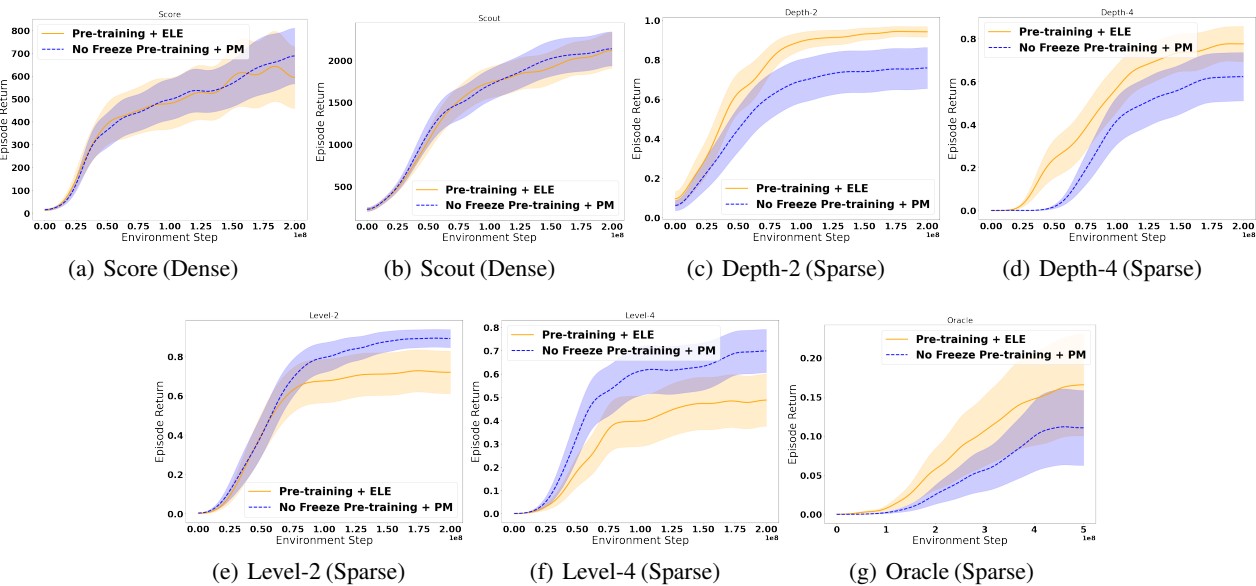

*Figure 9.* Episode returns of with/without freezing representations of contrastive pre-training during online phase. There is no significant difference with/without freezing representations of encoder during online training for most of the tasks. We report all other results with freezing the representations from the pre-training phase . All curves are reported over 5 random seeds ± one standard deviation.

## A.4. Implementation details

Hyperparameters for our approach are shown in Table 1. All other hyperparameters are identical to ELE (Anonymous, 2023), and the architecture of the encoder is the same as the one used in ELE. Each experiment was run on 8 TPUv3 accelerators using a podracer configuration (Hessel et al., 2021b).

| Hyperparameter | Value |
|---|---|
| Contrastive discount $\gamma$ | 0.95 |
| Batch size | 576 sequences |

*Table 1.* Muesli hyperparameters that differ between our approach and ELE.