# OpenReview forum: "Accelerating exploration and representation learning with offline pre-training"
_ICML.cc/2023/Workshop/ILHF — ILHF Workshop ICML 2023_

### Official Review · Reviewer_XmgU · 2023-06-03

**Rating:** 7
**Confidence:** 3

**Review:**

The hypothesis explored by the author suggests that enhancing exploration and representation learning can be achieved by training two separate models from a single offline dataset. The findings demonstrate that by employing noise-contrastive estimation to learn state representation and an auxiliary reward model independently from a collection of human demonstrations, significant improvements in sample efficiency can be achieved on the demanding NetHack benchmark.

The idea appears to be novel, and the experimental results are solid. Therefore, I would like to recommend for acceptance.

---

### Official Review · Reviewer_KbbP · 2023-06-17
**Worth Including in the Workshop**

**Rating:** 7
**Confidence:** 4

**Review:**

SUMMARY:
The authors demonstrate how offline data can be used to pre-train representations that, in turn, reduce the sample complexity of some state-of-the-art algorithms in the NetHack environment. Specifically, they learn two separate representations (one for exploration and one for representation learning) and combine them with Musli and their own Explore-Like-Experts algorithm to produce vastly improved empirical sample complexities in some NetHack subtasks.

COMMENTS:
I'm not sure if the conclusions of this work are that interesting for the community as they seem pretty self-evident (I would be surprised if pre-training didn't help) and are not that original outside of the ELE algorithm or the NetHack benchmark. However, it is hard to deny that the work does reach what could be a potentially useful conclusion in the context of future work and does manage to take a step forwards on this challenging benchmark. Those conclusions are well-supported by strong experiments in which I did not notice any glaring technical issues. The paper itself is also quite well-written, though it could do with some additional polish to the language in the earlier sections.

CONCLUSION:
I believe this work is of sufficient strength to justify acceptance to the workshop. I believe it's sufficiently relevant to the workshop to justify inclusion. On the latter point---and the latter point alone---, I would ask the meta-reviewer to weigh the opinion of the other reviewers more highly than my own.

---

### Decision · Program_Chairs · 2023-06-20

Accept